# The Relationship Between Psychophysiological and Psychological Parameters of Job Stress and Working Capacity of Loggers During the Fly-In Period

**DOI:** 10.3390/healthcare13182260

**Published:** 2025-09-09

**Authors:** Yana Korneeva, Natalia Simonova

**Affiliations:** 1Department of Psychology, Northern (Arctic) Federal University named after M.V. Lomonosov, Arkhangelsk 163002, Russia; n23117@mail.ru; 2Department of Labor Psychology and Engineering Psychology, Faculty of Psychology, Moscow State University named after M.V. Lomonosov, Moscow 125009, Russia

**Keywords:** job stress, working capacity, psychophysiological stress parameters, psychological stress parameters, forest harvesting, fly-in-fly-out method, shift work

## Abstract

**Background**: Scientific research on fly-in/fly-out (FIFO) workers has identified a gap in understanding the dynamics of job stress parameters among forest workers throughout the shift cycle. **Methods**: This study investigated the relationship between psychological and psychophysiological parameters of job stress and work capacity among loggers. The research was conducted during two simultaneous scientific expeditions in July 2024, involving 47 loggers from two teams with differing socio-psychological characteristics. Data were collected daily (morning and evening) using a battery of psychophysiological and psychological tests. Teams’ socio-psychological characteristics were assessed five times during the 15-day fly-in period. **Results**: The adaptation (beginning) and fatigue (end) phases of the shift were significantly more stressful than the middle period. During these critical phases, assessments of functional state showed greater consistency but were less favorable. Key findings indicate a psychological mobilization effect at the period’s start, where high subjective comfort coexisted with physiological strain. By the end, functional capabilities were maintained despite high fatigue. Furthermore, loggers in teams with a positive socio-psychological climate exhibited a more favorable functional state throughout the shift. **Conclusions**: The study’s novelty lies in its comprehensive mapping of the dynamic interplay between job stress and work capacity across the FIFO cycle, using both instrumental and questionnaire-based methods. The results underscore the critical influence of the team’s socio-psychological climate on worker well-being and highlight specific high-stress phases that warrant targeted interventions.

## 1. Introduction

Logging is considered to be a dangerous profession due to its specific nature. Physical (low temperatures, noise, and vibration) and production factors (heavy physical labor and risk of injury) in combination with organizational (disruption of biorhythms, intensity of workload, and duration of shift period) and psychological (psychosocial stress and group isolation) factors create a unique set of professional risks for loggers. One of the unfavorable factors is working outdoors, when the logger is exposed to a variety of weather conditions, depending on the season and prevailing temperatures: Weather conditions can complicate the performance of professional duties, for example, in rain, strong winds, extremely low temperatures in winter, and high temperatures in summer [1]. Noise and vibration are also among the production factors of loggers, which can slow down and complicate mental processes, reducing concentration, which contributes to inaccurate performance of actions [1,2,3]. The introduction of harvester logging has not only improved safety and reduced the accident rate but has also led to a risk of musculoskeletal disorders and increased psychological stress [4], as well as increased concentration and attention of operators over a long period of time [5]. The factors that make harvesters’ work difficult include heavy lifting, prolonged awkward postures in the lower back, repetitive movements, and prolonged periods of time in the same work breaks, which have a negative impact on the musculoskeletal system [6,7]. The logging process is complicated by the following professional environment factors: weather conditions, swampy terrain, great remoteness of forest plots (which requires more time for daily delivery of workers to the workplace), insufficient productivity and equipment breakdowns, lack of personnel, etc. [6,8].

Due to the remoteness of forest plots from populated areas (more than 100 km) and their constant movement through forest areas, logging enterprises use the shift work organization method or FIFO [8]. A fly-in-fly-out basis (FIFO) is defined as “any employment in which work is so isolated from workers’ homes that they are provided with food and housing in the workplace, and schedules are created whereby workers spend a fixed number of days at the field followed by a fixed number of days at home” [9] (p. 2). In FIFO work in logging, the shift duration is usually 12 h and employees work daily for 15 days without days off, followed by a 15-day rest fly-out period and then another 15-day fly-in period. This work involves 12 h day and night shifts, so in the middle of the fly-in period, there is a changeover, when day shift workers switch to night shift workers, and vice versa. The FIFO method is widely used in the extractive industries (oil and gas, diamond and mining, etc.), which extract minerals in remote areas with harsh climatic conditions of the Arctic and High North [10]. Long working hours (more than 12) and work requirements increase the likelihood of fatigue and stress in loggers [11]. Literature data show that working conditions and professional workloads have a negative impact on the functional state of loggers: There is an increase in stress, anxiety, as well as tension in different parts of the body (back, neck, etc.) [8,12,13]. The main occupational groups in logging enterprises are logging equipment operators (harvesters and forwarders) and truck drivers (for timber removal and dump truck drivers) [8]. The main factors affecting the work of truck drivers are long periods of intense concentration, as well as quick reactions, severe stress, and high long-term muscle load [8,14]. This leads to adverse effects on the health and psyche of workers and indirectly affects their work productivity [15,16]. Logging equipment operators have to work in uncomfortable positions, which leads to overexertion. The main problems of forest machine operators range from the difficulty of maneuvering machines on uneven terrain to variable soil conditions, which affect their productivity [17]. Increased mental workload is not only associated with discomfort but can also affect workers’ health and safety: Operators with severe fatigue are more likely to make mistakes, and this can lead to injuries or damage [18].

Loggers carry out their professional activities in conditions of group isolation in small teams (up to 30 people) during a 15-day fly-in period of daily 12 h work, followed by a rest period of the same duration. The successful performance of professional activities in shift work largely depends on maintaining group cohesion [19]. Teams need to coordinate their actions, communicate, and collaborate, relying on each other to successfully cope with external challenges [20]. O. Brown (2019) monitored changes in expedition team cohesion over time [21]. Isolation, coupled with intense physical exertion, makes it difficult to create and maintain effective teamwork, increases the likelihood of social conflicts, and emphasizes individual differences [22,23]. Expeditionary conditions are also characterized by monotonous daily tasks [24], which increases the risk of conflicts because loggers have to cope with boredom. The paper also provides data from other studies. In a 12-person group that completed a 61-day trek across Alaska, a positive relationship was found between cohesion and communication, perceived fairness in task distribution, and perceived quality of group leader decisions [24].

A previous study found that there is a discrepancy in the indicators of psychological and psychophysiological parameters of the functional states of loggers during a 15-day fly-in period [25]. According to psychophysiological parameters, throughout the entire fly-in period, employees experienced stress levels above the norm, with the highest risk occurring during the first four days of the shift, as well as the mid-shift changeover period (transition from night to day shift and vice versa) and the end of the fly-in period, during which a reduced level of performance and an acceptable condition level were observed [25]. At the same time, according to psychological parameters, positive dynamics of employees’ performance and well-being were established [25]. The established differences require clarification of the relationships between the psychophysiological and psychological variables describing the functional states of logging company workers at the beginning, middle, and end of the fly-in period. Thus, in modern science, there remains a significant gap in understanding the dynamics of the relationship between psychophysiological markers of job stress and the working capacity level of workers in the logging industry at various stages of the fly-in period.

Our previous studies have used biochemical, psychophysiological, and psychological methods to identify differences in the dynamics of working capacity and stress between two worker groups: FIFO employees in the Arctic oil and gas sector [26] and logging workers in the Far North [25]. These data are also confirmed by a number of studies that established differences in the physiological and perceived levels of stress in shift workers [27,28,29,30]. The established differences require clarification of how the various parameters of the functional states of workers at a logging enterprise are related at the beginning, middle, and end of the fly-in period. Our present study is relevant because, for the first time, the assessment of states was carried out in the summer period, when studying the relationships; data from daily double measurements of the states of workers were taken into account, using new, previously not used on similar samples, psychological testing methods for assessing functional states. The research was conducted in the summer period due to the need to study the impact of a complex of specific factors on workers: maximum work intensity, abnormally high temperatures for the North, increased activity of blood-sucking insects (gnats), and the polar day, which can provoke a unique pattern of job stress and decreased working capacity that is not typical for other seasons.

The aim of this study is to explore the complex interplay between psychological and psychophysiological markers of job stress and work capacity among FIFO loggers in the Arctic region. A key focus is to analyze how this interplay evolves during the fly-in cycle and is modulated by the socio-psychological climate of the work teams.

Research objectives:To carry out the comparative analysis of the socio-psychological characteristics of teams of two logging divisions.To identify and describe the features of job stress and the working capacity of FIFO workers of a logging enterprise of various divisions at the beginning, middle, and end of the fly-in period.To establish and describe the relationship between psychophysiological and psychological parameters of job stress and working capacity of FIFO logging workers at the beginning, middle, and end of the fly-in period.

Hypotheses:

Main hypothesis: Due to a much more stressful character of the adaptation and final stages of FIFO work, greater consistency of various assessments of job stress and working capacity of loggers is expected compared to the more favorable period of the middle of the fly-in period. These differences may indicate intersystemic consolidation of various adaptation methods in more severe conditions and their measured, distributed use in relatively favorable periods.

The special hypothesis posits that positive socio-psychological characteristics within a logging team are positively associated with dynamic changes in work capacity and job stress parameters among workers during the FIFO rotation period.

This study investigated the relationship between psychophysiological and psychological parameters of job stress and the working capacity of loggers at different stages of the fly-in period. Its findings hold significant value for public health and occupational medicine. By identifying the most demanding stages of the fly-in cycle, targeted preventive interventions can be implemented. These may include enhanced monitoring of stress and working capacity, along with psychological support.

The combination of psychophysiological and psychological markers we identified can form the basis for diagnostic panels or screening programs. This will enable on-site medical personnel (paramedics and physicians) to detect at-risk workers earlier, before the clinical onset of chronic stress, cardiovascular pathologies (e.g., arterial hypertension), and neuropsychiatric disorders.

As logging is a high-risk industry for injuries and accidents—often caused by human factors like fatigue, inattention, and emotional instability—applying our findings to the occupational safety system will help:− reduce industrial injuries through proactive management of worker fatigue and stress;− minimize economic losses related to absenteeism, sick pay, and employee turnover;− increase productivity by maintaining an optimal level of worker performance throughout the fly-in period.

## 2. Materials and Methods

The study is longitudinal.

### 2.1. Procedure

The study was conducted during two scientific expeditions to two separate divisions of a logging enterprise at forest plots (Karpogory village (geographical coordinates 64°00′02″ N, 44°26′42″ E) and Yasny settlement (64°02′08″ N, 44°04′46″ E), Arkhangelsk region of the Russian Federation) simultaneously from July 16 to July 31, 2024 during the entire 15-day fly-in period with daily double measurements (morning and evening) of occupational stress parameters. All the participants took part in the study on a voluntary basis after signing a written informed consent. All surveyed workers live in the Arkhangelsk region. Work and rest schedule: 12 h\12 h during the entire fly-in period without days off.

Due to the fact that both plots are located next to each other, have similar climatic conditions, and belong to the same organization, we believe the differences between them can be observed in the socio-psychological characteristics of the teams, which, under conditions of group isolation with the FIFO work method, can affect the professional stress and performance of workers. In this regard, these characteristics were assessed and taken into account in the study.

Testing the hypothesis about the relationship between various parameters of job stress and the working capacity of employees at various stages of the fly-in period was carried out on a common sample in order to identify general patterns.

### 2.2. Sample

The study involved 47 loggers aged 26 to 60 years. The sample is representative of the general population of loggers in the Arkhangelsk region of the Russian Federation due to the correspondence of the demographic indicators of the sample to the demographic indicators of the enterprise as a whole and the presence of respondents of different ages, education, experience, positions, and structural separate divisions (Table 1).

This longitudinal study was conducted via expeditions, where the researchers resided on-site for the entire fly-in period to conduct twice-daily monitoring of the loggers’ conditions. To maximize the sample size, the largest available plots were selected (approximately 30 loggers each). While a smaller plot of 12 loggers was also included, it required the same extensive expeditionary resources. The professional activities and organizational parameters were identical across both samples; the only geographical difference was their 30 km separation, as season and weather conditions were the same. Consequently, the samples were considered equivalent and were pooled to increase the overall sample size. The only objective, uncontrolled differences between the groups were in age and work experience. The study’s sufficient statistical power is supported by its longitudinal design and the high number of repeated measurements. The final compared sample sizes for stress and performance measurements were 161 (beginning), 194 (middle), and 164 (end of the fly-in period).

The study focused on a primary professional group: operators of logging equipment. This group comprised several specialized roles:

Harvester and forwarder operators: These operators work in tandem. Harvester operators perform felling, limbing, and cross-cutting using automated systems. Forwarder operators are then responsible for collecting, loading, and transporting the timber to the roadside using a crane-manipulator, ensuring the harvester’s continuous operation.

Support operators: This category included motor grader operators, who build and maintain the road network, and drivers who operate timber and dump trucks.

Technical maintenance: Specialists such as welders were included for their role in performing on-site minor equipment repairs.

All the loggers in the sample live in the Arkhangelsk region, get to the gathering place (v. Karpogory, s. Yasny) by car or rail, and from the gathering place to the shift camp, they are delivered by bus along a 100 km forest road. In the shift camp, workers live in cabins designed for 4 people, equipped with sleeping places, places for eating and resting, and a stove for heating. On the territory of the shift camp, there is a canteen and a bathhouse.

### 2.3. Methods

The diagnostic program was developed over a long period of time in the course of a number of studies [25,26,31], and it is described in detail and tested in other studies, including a multi-component assessment of conditions using hardware, survey, and projective methods for the most complete description of job stress and working capacity.

The functional states of loggers were assessed throughout the entire fly-in period using the following methods:(1)psychophysiological hardware techniques:

Variational cardiointervalometry (VCM) was performed using the UPFT-1/30 “Psychophysiologist” device [32]. This technique assesses an individual’s functional state by analyzing 128 cardiac cycles, recording the ECG signal, RR interval times, and their standard deviation.

The VCM methodology distinguishes six levels of functional state:

Optimal (6): Characterized by the ideal tension of regulatory systems required to maintain the body’s active equilibrium with the external environment.

Close to Optimal (5): A state of moderate tension where the body must utilize additional functional reserves to adapt to environmental conditions.

Permissible (4): A state of pronounced tension, involving the active mobilization of protective mechanisms.

Maximum Permissible (3): A state of regulatory system overstrain, marked by the insufficiency of protective and adaptive mechanisms and an inadequate response to environmental factors.

Negative (2): A state of regulatory system exhaustion, where the activity of control mechanisms declines. Individuals in this state must suppress their need for rest by willpower.

Critical (1): Characterized by functional disorders and a powerful, compelling need for rest as the restorative systems are activated.

− methods for assessing the characteristics of the cardiovascular system using the AngioCode mobile health tracker (ZMT LLC, Izhevsk, Russia) [33]. This device is included in the program for assessing the functional state of workers due to its active use at various enterprises by psychological services and labor protection services of industrial enterprises.− assessment of arterial pressure and heart rate (HR) using a tonometer with subsequent calculation of the coefficients [34].

Endurance coefficient (EC) according to the formula of A. Kvaas:EC = HR × 10/(SBP − DBP)

HR—pulse, SBP—systolic blood pressure, DBP—diastolic blood pressure. An increase in this indicator indicates a weakening, and a decrease indicates an increase in the functional capabilities of the cardiovascular system. The norm is from 12 to 15.

Adaptation potential (AP) or the index of the circulatory system functional state:AP = 0.011 × HR + 0.014 × SBP + 0.008 × DBP + 0.014 × B + 0.009 × BM − 0.009 × P − 0.27
where HR is pulse in beats per minute, SBP is systolic blood pressure in mm Hg, DBP is diastolic blood pressure in mm Hg, B is age in years, P is height in cm, and BM is body weight in kg.

Adaptation state assessment: Satisfactory state AP < 2.1; tense adaptation AP 2.1–3.3; unsatisfactory adaptation AP 3.31–4.3; adaptation failure AP > 4.3.

E.A. Pirogova’s physical condition index (PCI):700 − 3 × HR − 0.8333 × SBP − 1.6667 × DBP − 2.7 × B + 0.28 × BMPCI = 350 − 2.6 × A + 0.21 × H
where HR is pulse in beats/min, SBP is systolic blood pressure in mmHg, DBP is diastolic blood pressure in mmHg, Age in years, His height in cm, and BM is body weight in kg.

The following PCI standards can be used as gradations: Low—less than 0.375; Below average—from 0.375 to 0.525; Average—from 0.526 to 0.675; Above average—from 0.676 to 0.825; High—from 0.826 and more.

(2)psychological methods:

“State Scale” by E. Grol, M. Heider, adapted by A. B. Leonova [35,36,37] consists of 10 polar statements; as a result, the level of subjective comfort is calculated: 62 and more—high; 54–61—increased; 48–53—acceptable; 41–47—decreased; below 41—low.

The short eight-color Luscher test [38] was applied using the interpretive coefficients developed by G.A. Aminev [39]. All coefficients were calculated according to specific formulas that reflect particular combinations of the chosen colors. A more detailed description of each parameter and its calculation, based on the respondent’s color selections, is provided in our previous work [26].

According to M. Luscher, high performance is indicated by the placement of the green, red, and yellow colors in the leading positions of the sequence. This parameter is captured by Aminev’s performance coefficient, which ranges from 9.1 to 20.9. Luscher theory suggests that stress is indicated by the presence of primary active colors in the last positions of the subject’s ranking, as well as by the placement of brown, black, and grey cards in the first positions. This parameter ranges from 0 to 41.8.

Vegetative balance is calculated by comparing the positions of red and yellow colors against blue and green at the beginning or end of the sequence. It reflects the relationship between the sympathetic and parasympathetic nervous systems, i.e., between the activating and inhibitory processes in the body.

Personal balance is derived from the ratio of violet and brown to green and blue at the extremes of the sequence and indicates the relationship between internal (subjective) and external (objective) factors in an individual’s behavior. It reveals the degree to which a person is focused on their internal experiences versus external circumstances.

Heteronomy signifies the extent to which an individual is influenced by the external environment, social norms, or the opinions of others.

Aminev’s interpretation coefficients have been widely used in empirical studies and demonstrate good correlation with other parameters of human functional states [8,26].

The method of differentiated assessment of states of reduced performance (DASRWC) is by A.B. Leonova and S.B. Velichkovskaya. The method is an adaptation of the BMSII test by H. Plath and R. Richter [40,41]. The test comprises 40 questions. Responses are assessed on a 4-point scale (where 1 is “almost never” and 4 is “almost always”) to identify four severity levels of symptoms for the following mental states: monotony, satiety, stress, and fatigue.

Fatigue is characterized by exhaustion and impaired coordination of core cognitive processes. It results from prolonged and intense workload under a dominant motivation to cease work and rest. Score interpretation: ≤15 (low); 16–25 (moderate); 26–31 (pronounced); >31 (high).

Monotony is a state of reduced conscious control over task performance. It occurs during monotonous work involving frequent repetition of stereotypical actions in an impoverished external environment and is accompanied by feelings of boredom, drowsiness, and a strong desire to change activities. Score interpretation: ≤15 (low); 16–25 (moderate); 26–30 (pronounced); >30 (high).

Satiety (or mental satiety) is a state of aversion to overly simple, subjectively uninteresting, or meaningless work. It manifests as a strong urge to either stop the activity altogether or to introduce variety into the established routine. Score interpretation: ≤16 (low); 17–24 (moderate); 25–30 (pronounced); >30 (high).

Stress (or tension) is a state of heightened mobilization of psychological and energetic resources in response to increased task difficulty or subjective significance. Score interpretation: ≤16 (low); 17–24 (moderate); 25–30 (severe); >30 (high).

Depression, Anxiety, and Stress Scale (DASS-21) by S. Lovibond and P. Lovibond, adapted by A.A. Zolotareva [42,43,44,45], consists of 21 statements to be rated on a 4-point scale from never to almost always. The depression scale measures such mental states and experiences as dysphoria, hopelessness, devaluation of life, self-abasement, and apathy (1–8 points indicate experiences of depressive spectrum symptoms from mild depression (subdepression) to moderate; more than 8 points indicate high severity of depressive state). The anxiety scale assesses vegetative arousal, situational anxiety, and subjective experience of experiencing anxiety states (1–7 points—from mild to moderate anxiety; more than 7 points—high severity). The stress scale is sensitive to chronic non-specific arousal and diagnoses difficulties with relaxation, nervous arousal, and irritability (2–10 points—from mild to moderate stress, more than 10 points—severe stress).

Figure 1 shows the methods used to assess functional states and the parameters measured by them. Diagnostics of the functional states of loggers were carried out daily in the morning and evening using the VCM, CVMR, AngioCode, tonometer, state scale, M. Luscher test, and DASS21. The DASRWC method was used every three days due to the content of the questionnaire.

To describe the socio-psychological characteristics of the two units, the following methods were used five times during the fly-in period (every three days), and the socio-psychological climate method was used once on the 4th day of the fly-in period:(1)S.E. Seashore’s group cohesion [46,47]. Group cohesion was understood as an integral indicator reflecting the degree of emotional attractiveness of the group for its members, the strength of the participants’ desire to maintain membership in the team, and the level of unity in achieving common goals. This method allows us to assess the degree of integration of the group, its cohesion into a single whole. It consists of 5 questions, each of which has from 4 to 6 answer options. The final indicator can range from 5 (very unfavorable assessment) to 19 points (very high assessment).(2)F.E. Fiedler’s psychological atmosphere, adapted by Yu. L. Khanin [48,49,50]. The psychological atmosphere in a team was understood as a dynamic emotional-evaluative characteristic of interpersonal relations in a group, reflecting the subjective perception of comfort, trust, and general emotional background of interaction by participants. It is based on the semantic differential method. Employees give an assessment of the group according to the proposed bipolar scales, according to 10 dichotomies. The final indicator fluctuates from 10 (the most positive assessment) to 80 (the most negative). The lower the coefficient, the more favorable the assessment of the psychological atmosphere in the team.(3)V.A. Rozanova’s group motivation [51]. Group motivation is an integral indicator reflecting the degree of team members’ involvement, interest, and focus on achieving common goals. It is compiled according to the semantic differential type and contains 25 statements with a rating scale from 1 to 7 points. The results are assessed based on the sum of points indicated on the questionnaire form: 25–48 points—the group is negatively motivated; 49–74 points—the group is weakly motivated; 75–125 points—the group is insufficiently motivated to achieve results; 126–151 points—the group is sufficiently motivated to succeed in its activities; 152–175 points—the group is positively motivated to succeed in its activities.(4)Socio-psychological climate by O.S. Mikhalyuk and A.Yu. Shalyto [52]. Socio-psychological climate is a qualitative aspect of interpersonal relations in a group, manifested as a set of psychological conditions that contribute to or hinder productive joint activities and comprehensive development of the individual. The diagnostics of the socio-psychological climate is carried out according to three main parameters: the emotional component (satisfaction with relationships); the cognitive component (assessment of the business qualities of colleagues); the behavioral component (readiness for joint activities).

### 2.4. Data Analysis

The statistical processing of the results was carried out using descriptive statistics, Student’s *t*-test, Mann–Whitney’s U test, and correlation analysis using Spearman’s rank correlation coefficient with the help of the IBM SPSS Statistics 26.00 package. The permissible level of the first type of error took into account the multiplicity of hypotheses being tested.

When conducting the correlation analysis, only those factors were taken into account that had statistically significant relationships at a level of *p* less than 0.04 with the parameters of the functional states of workers (group assessment of the probability of error (I type) using the Bonferroni method modified by Holm).

For the purpose of the study, all professional stress measurements were divided into three groups depending on the stage of the fly-in period: (1) days 1–3—the beginning of the fly-in period (a total of 63 and 98 measurements on the first and second plots, respectively); (2) days 6–8—the middle of the fly-in period (79 and 115 measurements); (3) days 11–13—the end of the fly-in period (61 and 103 measurements). The introduction provides data on the previous expedition, where it was established that the most stressful days of the fly-in period are the beginning (1–3 days), the end (11–13), and the middle (6–8 days) due to the shift change. These data are also on other samples of FIFO workers in the oil and gas production industry in the Arctic [26].

Using Student’s *t*-test (under the assumptions of normal distributions and homogeneity of variances) and the Mann–Whitney U test (if it is necessary to use a non-parametric test), statistically significant differences in the socio-psychological characteristics of the teams (Table 2), psychophysiological and psychological parameters of the functional states of loggers in the two divisions (Table 3 and Table 4) were established.

## 3. Results

### 3.1. Socio-Psychological Characteristics of Teams of Two Divisions at the Beginning, Middle, and End of the Fly-In Period

Table 2 shows descriptive statistics and the level of significance of differences in the socio-psychological characteristics of the teams of the two divisions at different stages of the fly-in period.

According to Table 2, statistically significant differences between the subdivisions were established in group motivation in the middle of the fly-in period, which was higher among the employees of the plot in the village of Karpogory than in Yasny settlement. At the same time, both subdivisions were sufficiently motivated to work together. Group cohesion differed among the teams of the two subdivisions at the beginning and at the end of the fly-in period and was also higher in the subdivision in the village of Karpogory than in Yasny settlement. The psychological atmosphere in the teams differed at all stages of the fly-in period and was also higher in the subdivision in the village of Karpogory than in Yasny settlement (the lower the indicator, the more favorable the psychological atmosphere). Using Pearson’s *χ*^2^ for contingency tables, statistically significant differences were established in the distribution of employees of the two subdivisions with respect to the emotional and behavioral components of the socio-psychological climate (Figure 2). In the Karpogory subdivision, more than 94.7% of employees had a positive emotional attitude towards the team, and 52.6% expressed a positive attitude and readiness for joint activities, while in Yasny settlement, only 65.2% experienced positive emotions towards the team and 13% were ready to work in such a group.

Thus, based on the analysis of data on group motivation, cohesion, and socio-psychological climate in teams, we can conclude that the division of the village of Karpogory is a team with positive socio-psychological characteristics, while the Yasny team has moderate socio-psychological characteristics.

### 3.2. Peculiarities of Professional Stress in Loggers of Two Divisions at the Beginning, Middle, and End of the Fly-In Period

Table 3 and Table 4 provide descriptive statistics for those professional stress parameters for which statistically significant differences were established at different stages of the fly-in period at the beginning, middle, and end of the fly-in period.

According to the data in Table 3, Table 4 and Table 5, statistically significant differences were established in the parameters of the functional states of loggers of the two divisions, measured using the DASRWC, DASS-21, CVMR, VCM methods, as well as using the AngioCode device and the coefficients based on arterial pressure.

According to Table 3, the stress level (according to the Angiocode device) is lower among the Karpogory employees at the beginning of the fly-in period and higher in the middle and at the end of the fly-in period, in comparison with the Yasny team (while their values are below average). Systolic pressure was higher among the Yasny loggers at all stages of the fly-in period. The endurance coefficient (according to Kvaas) corresponded to the norm among the members of Yasny settlement during the entire fly-in period, and among the Karpogory employees, it was slightly higher, which indicates a weakening of the functional capabilities of the cardiovascular system. The adaptive potential was reduced among the employees of both divisions during the fly-in period, while Yasny was characterized by more intense adaptation. The physical condition indices of the Karpogory loggers were below average during the entire fly-in period, and in Yasny settlement, they were average. However, at the end of the fly-in period, they were above average. The Karpogory workers demonstrated above-average speed in performing a complex visual-motor reaction and were characterized by above-average operator performance throughout the entire fly-in period, while the representatives of Yasny settlement performed the complex visual-motor reaction more slowly, and their operator performance was below average. The level of functional capabilities of the Karpogory loggers was high, and their functional state was acceptable throughout the entire fly-in period. At the same time, the Yasny loggers had an average level of functional reserves of the cardiovascular system and a more favorable functional state level.

Thus, according to the psychophysiological parameters measured using a tonometer and the coefficients calculated on its basis, the Karpogory loggers had less favorable conditions, characterized by weakened endurance and an average index of physical condition; however, according to other methods (CVMR, VCM, AngioCode)—more positive indicators: above average speed of visual-motor reactions, average operator performance, above average level of functional capabilities, and low stress level. The Yasny employees, on the contrary, had a higher endurance (average level), an average level of physical condition, but they felt tense adaptation more acutely, their speed and operator performance were below average, and their level of functional capabilities of the cardiovascular system was average.

The employees of the Karpogory subdivision were characterized by moderate fatigue, satiety, and monotony, which were higher than those of the representatives of the other subdivisions. The stress level measured by DASRWC and DASS-21 was characterized by higher values for the employees of the Yasny subdivision. Depressive and anxious states were uncharacteristic for the Karpogory loggers, while in their colleagues from Yasny, they were manifested only slightly. It should be noted that despite higher professional stress, the employees of Yasny showed higher labor productivity—in the middle and at the end of the fly-in period (Table 5), they statistically harvested more wood than their colleagues from Karpogory.

### 3.3. The Relationship Between Psychophysiological and Psychological Parameters of Professional Stress in Loggers in the Far North at the Beginning, Middle, and End of the Fly-In Period

At the next stage of the study, to identify the relationship between the psychophysiological and psychological parameters of professional stress in loggers in the Far North at the beginning, middle, and end of the fly-in period, we used correlation analysis (the Spearman coefficient). Based on its results, three correlation pleiades were compiled for those parameters that have statistically significant relationships (Figure 3, Figure 4 and Figure 5). In the correlation pleiad, only those factors are noted that have statistically significant relationships with a strength of 0.3 and at *p* ≤ 0.001 (group error probability assessment (I type) according to the Bonferroni method in the Holm modification).

As can be seen in Figure 3, a greater number of connections at the beginning of the fly-in period were established between the psychophysiological parameters of the CVMR method and the psychological parameters of the DASRWC, DASS21, and subjective comfort methods. At the same time, opposite trends were revealed: The more unfavorable the functional state (according to the VCM) is according to the hardware characteristics, the higher the workers rated their subjective comfort, and the higher their operator performance was according to the CVMR. This indicates, on the one hand, a compensatory reaction (increased performance due to a positive psychological attitude), and, on the other hand, the inadequacy of using only subjective-evaluation methods when monitoring the state. Employees strive to perform their work efficiently even with a decrease in their internal resources, which is explained by the peculiarity of the polar tension syndrome, when unfavorable changes are realized later than they manifest themselves at the body level [25]. The agreement in the assessments of the state according to the DASRWC questionnaire with the hardware method of the CVMR and the parameters measured based on arterial pressure indicates its prognostic value. It is interesting to note that the higher the level of functional capabilities and functional state according to the VCM, the more pronounced the satiety in workers. Also, the more dominant the type of wave C according to AngioCode, which is more typical for young people under 30, the more pronounced the monotony.

With sufficient internal resources at the beginning of the fly-in period, monotony and satiety develop more intensively. Perhaps, these employees need variety in work, which is impossible to implement in these work positions.

In the middle of the fly-in period (Figure 4), statistically significant relationships were established between the parameters of the DASRWC and CVMR methods: The higher the stress, the lower the operator performance, the higher the satiety, the more errors in performing CVMR. The lower the level of functional capabilities according to the VCM, the more anxiety, stress, and depressive states were manifested according to DASS21. This indicates consistency in the assessments of the psychological and psychophysiological parameters of the functional states of employees. At the same time, the higher the level of functional capabilities, the more fatigue is expressed in loggers. As indicated earlier, in the middle of the fly-in period, there is a shift change, which is an additional stress factor. However, if there are internal reserves, workers cope with this, experiencing fatigue, but an insufficient level of resources leads to the development of stress, anxiety, and depressive states during this period.

According to the data in Figure 5, at the end of the fly-in period, as well as at the beginning, a greater number of statistically significant relationships were established between the psychological and psychophysiological parameters of the functional state of loggers. Significant relationships were found between the parameters measured using the DASRWC method and the VCM, CVMR, and Angiocode methods. The higher the monotony, the higher the stress level and the more elastic the vessels (Angiocode); the higher the fatigue, the lower the level of sensorimotor reactions; the higher the stress, the lower the operator performance of loggers at the end of the fly-in period; the lower the level of functional capabilities, the higher the anxiety. The opposite trend was found between the parameters of M. Luscher’s and VCM methods: The lower the level of functional capabilities, the higher the performance (according to the projective indicator). At the beginning of the fly-in period, such trends indicate the active inclusion of mechanisms of psychological regulation of functional states. The results obtained are partially consistent with foreign researchers who found that levels of perceived stress are not associated with levels of physiological stress [8,53]. Hjortskov and colleagues found that at low and medium levels of perceived stress, physiological stress may not occur, and when perceived stress is high, an increase in physiological stress may be observed [53].

## 4. Discussion

The results of this study made it possible to determine the specifics of the functional states of loggers at different stages of the summer fly-in period, which are consistent with the data obtained in another expedition in December 2020 at a forest plot near the village of Karpogory [25]. The following general trends were established: Throughout the fly-in period, favorable functional states and a high level of functional capabilities were observed (in this study, workers had a close to optimal state and above average functional capabilities); operator performance (OP) below average; high performance and low stress (according to M. Luscher’s method and G.A. Aminev’s coefficients); well-being, activity and mood (WAM) at a high level (in this study, subjective comfort at a moderate level). At the same time, a number of authors have found that logging is more difficult and less productive in summer than in winter. One of the reasons for this is hot weather (from 35° to 37 °C), which contributes to overheating of equipment and more frequent cleaning of air filters when working near dusty roads, as well as more equipment problems. Hot weather leads to increased operator fatigue, which affects their productivity [54]. In addition, in summer, trees need to be passed 4 or 5 times through the harvester head for better debarking of logs, and in winter, 1–2 passes are enough. Operator fatigue and equipment problems in summer are increased by long shifts, which leads to a decrease in average productivity per shift of up to 34% [54]. E. Lagerstrom et al. found that there are more accidents in summer [55]. The present study did not establish such links regarding more unfavorable conditions in the summer compared to the winter. Numerous studies have found that levels of perceived stress are not related to levels of physiological stress [56,57], etc. Hjortskov et al. believed that average levels of perceived stress may not cause a response in the form of physiological stress. However, when perceived stress is high, this can contribute to an increase in physiological stress [53]. Other authors have also found that people with low levels of perceived stress do not have a connection between perceived and physiological stress levels [56,57].

The form of perceived stress is also important and can influence the level of physiological stress. Stress experiences associated with uncertainty, novelty, distress, anxiety, helplessness, or lack of control trigger physiological stress reactions more often than other forms of stress (e.g., habitual or ordinary stress) [53,58]. The relationships between age and stress level, as well as length of service and fatigue in employees, obtained in the study, are consistent with the results of other authors. A study by H. Kymalainen et al. found a decrease in performance with age in forest machine operators in Finland [59]. In the present study, the beginning and end of the fly-in period were characterized by a higher level of stress reaction in older forest harvesters. Moreover, psychological methods demonstrate a connection with the length of service of workers only at the beginning of the fly-in period, which, in our opinion, is associated with an orienting reaction—an attempt to assess the upcoming difficulties and prepare to overcome them. This is not true for less experienced loggers because of their lack of a differentiated image of the labor object during the fly-in period. The influence of experience on the functional state of employees demonstrates the opposite nature of the relationship between stress and age in the middle of the fly-in period, which confirms the completion of the orienting stage for professionals, in contrast to beginners, against the background of resources that have not yet been spent. The hypothesis of the study was confirmed: Due to the greater stressfulness of the adaptation and completion stages of the fly-in period, a greater consistency of various assessments of the functional states of loggers is expected compared to the more favorable period in the middle of the fly-in period. Consistent relationships between objective (CVMR) and projective (the Luscher method) parameters were established, as well as multidirectional relationships between subjective comfort and objective parameters of the functional state of loggers, which is consistent with the results of our previous study conducted on a sample of FIFO workers in the oil and gas production industry in the Arctic [26]. In our previous study, the maximum number of relationships was established between objective indicators of cortisol in saliva (stress), CVMR indicators (operator working capacity), VCM (level of functionality), and interpretation coefficients (performance, stress, and vegetative balance) according to the Luscher’ test in FIFO personnel [26]. The present study has revealed a greater number of statistically significant relationships between the parameters of functional states measured using the following methods: (1) DASRWC, CVMR methods, and coefficients based on blood pressure measurements; (2) subjective comfort methods and parameters of the AngioCode device; (3) G.A. Aminev’s interpretation coefficients for the Luscher test and the CVMR method parameters.

The addition of the DASRWC, DASS21 questionnaires to the study design and the calculation of pressure-based coefficients made it possible to establish the specifics of changes in the functional states of FIFO workers of a logging company and use these data in practical work at enterprises. Due to the fact that blood pressure is measured during pre-shift examinations at logging sites, the calculation of additional indicators based on it and correlation of individual results with the obtained links with other parameters of workers’ states will allow giving feedback to workers about their condition and recommendations for its optimal maintenance during the fly-in period. The obtained reliable statistically significant links between the equipment parameters and the results of the DASRWC and DASS21 questionnaires substantiate the adequacy of their use separately from psychophysiological testing. Such comprehensive monitoring of the functional state of workers will ensure control and timely intervention of psychologists and labor physiologists in order to correct them for the purpose of maintaining the health and professional longevity of workers.

The findings of the study can be adapted to a wide range of occupations characterized by similar production factors (FIFO work in group isolation conditions, for example, the oil and gas industry, the mining industry). This opens up significant prospects for the development of universal protocols in the public health system.

Based on the findings of this study, the following practical recommendations are proposed to enhance performance and safeguard the health of loggers:For the occupational safety system and employers:
− Enhance safety protocols during critical periods: Strengthen oversight of occupational safety compliance during the beginning and end of the fly-in period. Temporarily restrict or increase supervision for the most hazardous tasks requiring high concentration (e.g., felling trees, working near machinery) during these high-risk phases.− Implement targeted psychophysiological relief programs:



Beginning of fly-in period: Focus on accelerated adaptation techniques and anxiety reduction (e.g., guided breathing exercises, adaptation briefings).

End of fly-in period: Introduce controlled recovery programs to combat cumulative fatigue (e.g., short relaxation sessions, facilitating quality rest).

− Modify work-rest schedules: consider adjusting shift structures, for example, by reducing shift duration to 11 h and increasing rest time. This could be achieved by logistical improvements, such as ensuring transportation from the shift camp to the work site takes no more than 20–30 min.

2.For the medical support and public health systems:

− Implement a functional state screening system:

Utilize the DASRWC questionnaire as the primary tool for weekly screening of all personnel, as its sufficiency and ease of use have been validated.

Conduct more in-depth psychophysiological testing (using hardware methods) selectively for workers in the “risk group” (as identified by DASRWC results) or during periodic medical examinations. This targeted approach optimizes the medical unit’s resources.

− Develop early intervention protocols for specialists: create standardized work protocols for psychologists and medical staff:

Beginning of fly-in period: techniques to reduce adaptation stress and foster team cohesion.

End of fly-in period: methods for providing emotional support and cognitive mobilization to ensure work is completed safely.

− Establish professional longevity programs: use the data to create individualized rehabilitation programs for the inter-shift period. These programs should aim to compensate for the identified loads (e.g., cardiovascular rehabilitation, stress management programs).− Create educational materials for workers: develop clear guides and memos that explain the typical dynamics of their condition during a shift, teach self-monitoring techniques, and emphasize the importance of seeking help during critical periods.

The hypothesis that dynamic shifts in the job stress and working capacity parameters of loggers during the fly-in period are positively associated with positive socio-psychological characteristics of the logging team was partially confirmed. Loggers working in a team with positive socio-psychological characteristics are distinguished by a more favorable level of functional state during the fly-in period. This can be explained by a number of factors identified in studies by other authors. Communication in a team is important not only in terms of productivity but also in terms of mental health. Team members experience greater stress when they cannot discuss personal problems with their peers [60]. Even when there are no overt conflicts, team performance may decline with prolonged exposure to the environment. Palinkas et al. in their study of multinational crews at Antarctic research stations found that the percentage and number of crew members seeking interaction declined over the course of the mission (8 months) [22]. A. G. Shchurov pointed out that the “Group Cohesion Index of K. E. Sishor’s” [61] method can be used to study a ship’s crew group cohesion. The study yielded data indicating a fairly high level of group cohesion. V. I. Lebedev interpreted group isolation as the forced presence of a group of people in a limited space, with a minimum of sensory stimuli and constant contact with the same people [62]. There are several negative impacts on members of such groups: The nervous system becomes weaker, which is expressed in increased irritability and emotional outbursts in conflict situations; exhaustion from lack of information; the need for constant self-control and concealment of true emotions due to the constant presence of other people; the feeling of loneliness caused by the inability to choose one’s social circle; the impact of work schedule at some enterprises on the general condition of employees. These features of social interaction can lead to unfavorable functional states and decreased work efficiency [62]. D. K. Sharipov conducted a study to assess the adaptation of the body in conditions of low temperature and the isolated environment of Antarctica [63].

The obtained results yield practical recommendations for the targeted management of the socio-psychological climate in logging teams as a resource for maintaining health and performance:−form stable work teams, considering employee preferences during staffing to enhance work comfort;−conduct an annual assessment of the socio-psychological climate, including employee feedback on improving intra-team and management interaction;−incorporate socio-psychological climate metrics into the key performance indicators (KPIs) for foremen and line managers to incentivize their focus on this issue;−hold regular short meetings (e.g., toolbox talks). Use these not only for safety briefings and tasks but also to openly discuss difficulties, assign tasks considering employee input, and express gratitude for good work. This fosters a greater sense of involvement and fairness.

Limitations of the study. The limitation of the study was the inclusion in the sample of the study of only representatives of three professional groups (logging equipment operators, truck drivers, and maintenance workers). The sample size (47 employees) is a limitation of the study that could be addressed by future studies with a similar design. Also, the study was conducted in one of the regions of the Russian Federation, which can be clarified when conducting research in other regions with different climatic conditions and terrain specifics. The results obtained are based on the methods of interviews, questionnaires, psychophysiological, and psychological testing. The use of other tests and methods may allow us to refine the results obtained.

Future research directions to address these limitations:Expanding the professional and geographic representativeness:

Conduct a multicenter study including employees from other key logging regions of Russia (e.g., the Far East, the Urals, the Northwest) to assess the impact of local specifics and climatic conditions (e.g., humidity, elevation changes, different daylight hours) on the results obtained.

Expand the professional composition of the sample by including representatives of other specialties that are critical to the production process—line managers (foremen), labor protection, and safety employees. This will provide a complete picture of the stress load at the enterprise.

2.Increasing the size and diversifying the sample:

Conduct a longitudinal cohort study involving a larger number of respondents (n > 100) to increase the statistical power of the analysis and ensure greater representativeness and the possibility of using multivariate analysis methods to identify hidden relationships.

Form a stratified sample, ensuring proportional representation of all professional groups, age cohorts, and workers with different lengths of FIFO work experience, to analyze the impact of these factors on the parameters under study.

3.Deepening the methodological design:

Implement a longitudinal design with multiple measurements not only during one shift but also over several FIFO cycles (including rest periods at home) to assess the cumulative effect and long-term consequences of job stress, as well as to study recovery processes.

## 5. Conclusions

This study established statistically significant relationships between the psychological and psychophysiological parameters of functional states in Arctic loggers across a 15-day fly-in period. The most significant findings are summarized below.

### 5.1. Dynamics of Functional State

The adaptation (beginning) and fatigue (end) phases of the fly-in period were identified as the most stressful. During these stages, assessments of functional state were more consistent but less favorable compared to the more stable middle phase. Notably, we observed a psychological mobilization effect at the start of the period, where subjective comfort was maintained despite physiological strain. Conversely, by the end, workers demonstrated a relative maintenance of functional capabilities against a backdrop of accumulating fatigue.

### 5.2. Practical Application of Monitoring

A strong correlation was found between objective (equipment-based) parameters and the subjective DASRWC questionnaire. This validates the use of this concise questionnaire as a standalone, efficient tool for routine monitoring of functional states in remote industrial settings, enabling timely psychological or physiological intervention.

### 5.3. Impact of Team Socio-Psychological Climate

The socio-psychological climate of a team was a critical factor in worker well-being: Positive climate teams: workers in these teams exhibited a more favorable functional state throughout the fly-in period, characterized by moderate fatigue and stress, average to above-average operational performance, and high adaptive and communicative abilities.

Moderate climate teams: workers in these teams reported higher stress, lower operational performance, and only an acceptable functional state. This appears linked to their lower adaptive abilities and communicative potential.

### 5.4. Performance Outcomes

A notable finding was that teams with a moderate socio-psychological climate achieved higher timber harvest volumes mid- and end-period. This may be driven by a stronger adherence to organizational norms and a heightened focus on meeting performance targets, even at a higher personal cost of stress.

The functional state of loggers is dynamic and significantly influenced by the phase of the fly-in period and the quality of their team environment. Implementing simplified monitoring tools like the DASRWC questionnaire can help safeguard worker health and promote professional longevity in this demanding industry.

## Figures and Tables

**Figure 1 healthcare-13-02260-f001:**
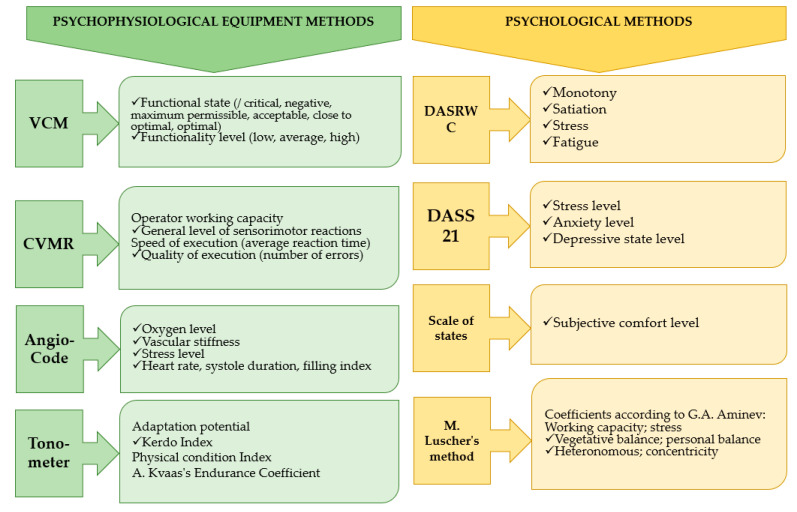
Methods for diagnosing functional states of loggers and parameters evaluated by them.

**Figure 2 healthcare-13-02260-f002:**
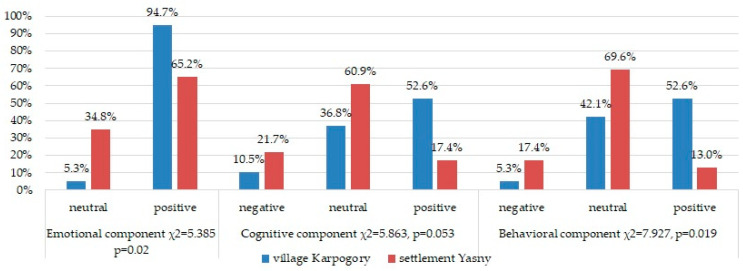
Characteristics of the socio-psychological climate in the teams of two divisions.

**Figure 3 healthcare-13-02260-f003:**
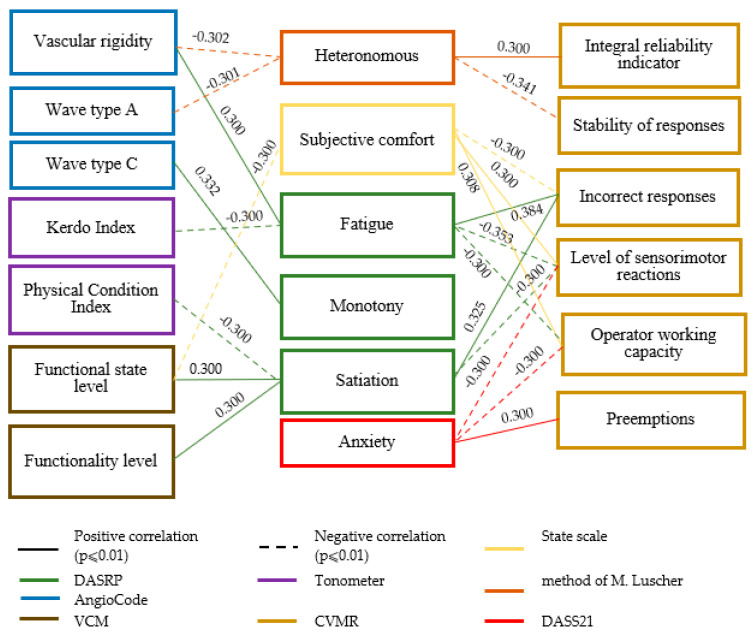
The relationship between psychological and psychophysiological parameters of the functional state of loggers at the beginning of a fly-in period (1–3 days, N = 160).

**Figure 4 healthcare-13-02260-f004:**
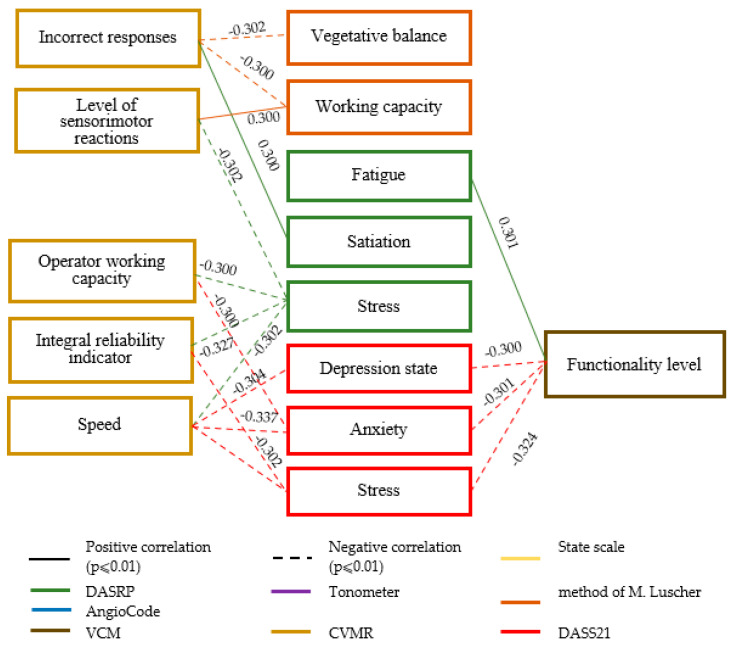
The relationship between psychological and psychophysiological parameters of the functional state of loggers at the middle of the fly-in period (6–8 days, N = 188).

**Figure 5 healthcare-13-02260-f005:**
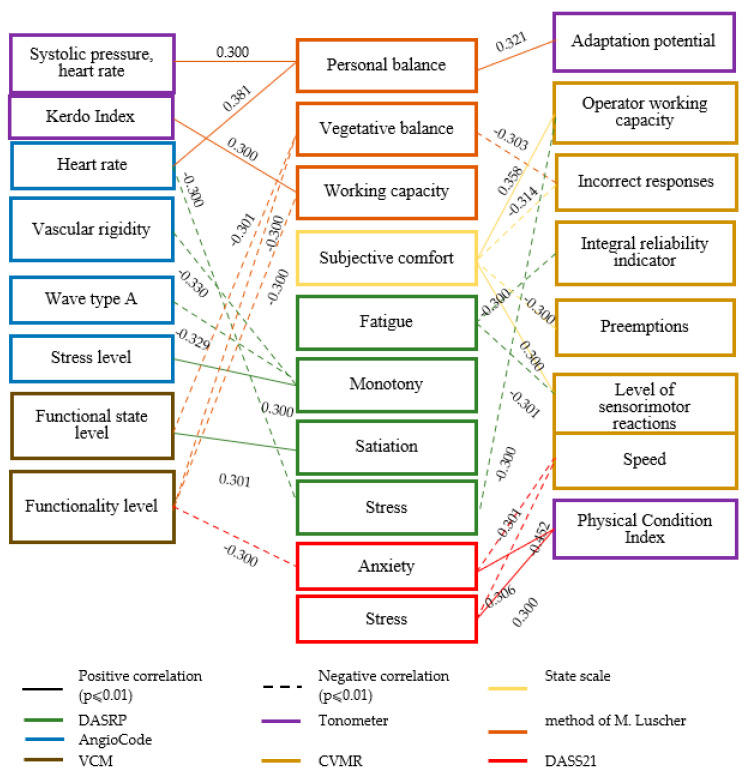
Interrelationship between psychological and psychophysiological parameters of the functional state of loggers at the end of the fly-in period (11–13 days, N = 164).

**Table 1 healthcare-13-02260-t001:** Demographic characteristics of the samples.

	v. Karpogory	s. Yasny
Number of participants	20	27
Age (average age)	26–59 years (38.26 ± 9350)	27–60 years (44.85 ± 9607)
Experience in FIFO work (average length of service)	1.5–25 years (8.26 ± 6701)	0.1–36 years (10.20 ± 9527)
Experience in position (average length of service)	0.1–18 years (5.40 ± 4765)	0.1–38 years (10.77 ± 12,166)
Education level	25% general secondary, 60% secondary vocational, 15% higher	45%—general secondary, 35%—secondary vocational, 20%—higher education
Positions	65% forestry machine operators (harvester, forwarder) 15% drivers 10% motor grader and excavator operators 10% maintenance specialists (welders, general workers)	37.04%—forestry machine operators (harvester, forwarder); 33.33%—drivers; 11.11%—machine operators (motor grader, excavator); 14.82%—maintenance specialists; 3.70%—foremen

**Table 2 healthcare-13-02260-t002:** Socio-psychological characteristics of the teams of the two divisions.

Parameter, Methodology	Beginning of Fly-In Period			Middle of Fly-In Period			End of Fly-In Period		
	K	Ya	*p*	K	Ya	*p*	K	Ya	*p*
Group motivation, V.A. Rozanova	142.8; 22.00	126.5; 21.86	0.057	139.3; 22.49	124.2; 18.89	0.047	135.3; 26.37	124.7; 17.98	0.204
Group cohesion, K.E. Seashore	14.9; 1.78	13.9; 2.45	0.026	14.3; 1.96	14.16; 2.39	0.947	15.0; 1.80	14.0; 1.86	0.001
Psychological atmosphere, F. Fiedler	17.7; 10.98	25.3; 10.38	0.001	17.9; 7.03	24.5; 9.7	0.001	19.4; 5.62	27.2; 7.91	0.001

Note: K—subdivision of the village of Karpogory, Ya—subdivision of Yasny settlement.

**Table 3 healthcare-13-02260-t003:** Psychophysiological parameters of professional stress and performance of loggers of the two divisions at the beginning, middle, and end of the fly-in period.

Condition Parameter, Methodology	Beginning of Fly-In Period (M; SD)			Middle of Fly-In Period (M; SD)			End of Fly-In Period (M; SD)		
	K N = 63	Ya N = 98	*p*	K N = 79	Ya N = 115	*p*	K N = 61	Ya N = 103	*p*
Stress level *	175.9; 269.19	180.9; 147.74	0.05	200.7; 238.01	165.2; 194.79	0.011	159.5; 200.24	111.2; 126.27	0.016
Systolic pressure **	131.9; 17.00	137; 17.56	-	127.6; 13.97	136.1; 17.71	0.001	127.4; 14.07	129.7; 15.97	-
Quasa endurance coefficient **	16.2; 4.51	14.5; 3.97	0.018	17.0; 4.52	14.9; 4.53	0.001	17.1; 4.61	15.3; 4.61	0.008
Adaptive potential **	2.8; 0.46	3.1; 0.35	0.001	2.7; 0.41	3.1; 0.36	0.001	2.7; 0.43	3.0; 0.37	0.001
Physical condition index **	0.5; 0.20	0.6; 0.14	0.001	0.5; 0.18	0.6; 0.15	0.001	0.5; 0.21	0.7; 0.17	0.001
Speed ***	4.3; 1.09	3.3; 0.98	0.001	4.4; 1.08	3.3; 1.09	-	4.6; 0.99	3.5; 0.97	0.001
Operator performance ***	3.3; 1.44	2.5; 0.95	0.001	3.5; 1.65	2.8; 0.89	0.003	3.8; 1.63	2.8; 0.88	0.001
Functional capacity level ****	5.2; 1.7	3.7; 1.3	0.001	5.5; 1.5	3.5; 1.26	0.001	5.0; 1.75	3.7; 1.46	0.001
Functional condition level ****	3.2; 1.33	4.2; 1.22	0.001	3.4; 1.23	4.0; 1.19	0.003	3.4; 1.07	4.3; 1.49	0.001

Note: K—Karpogory subdivision, Ya—Yasny subdivision; * AngioCode, ** coefficients according to tonometer data, *** CVMR; **** VCM.

**Table 4 healthcare-13-02260-t004:** Psychological parameters of job stress and working capacity of loggers of two divisions at the beginning, middle, and end of the fly-in period.

Condition Parameter, Methodology	Beginning of Fly-In Period (M; SD)			Middle of Fly-In Period (M; SD)			End of Fly-In Period (M; SD)		
	K N = 63	Ya N = 98	*p*	K N = 63	Ya N = 98	*p*	K N = 63	Ya N = 98	*p*
Vegetative balance *	4.8; 4.32	5.8; 3.75	-	3.8; 5.12	5.3; 3.87	-	2.9; 5.28	6.1; 3.22	0.001
Personality balance *	0.9; 2.62	1.7; 2.62	-	1.1; 2.89	1.5; 2.55	-	0.2; 3.35	1.6; 2.74	-
Heteronomousness *	−0.8; 2.93	−0.9; 2.62	-	−0.5; 2.58	−0.5; 2.67	-	−1.0; 3.05	−0.3; 2.46	-
Workability *	18.7; 2.42	19.1; 1.70	-	18.7; 2.51	18.7; 1.72	-	18.0; 2.83	19.0; 1.68	-
Stress *	6.6; 7.58	6.5; 6.07	-	5.5; 6.67	5.7; 6.35	-	6.3; 7.30	6.8; 7.53	-
Subjective comfort **	53.8; 8.88	52.8; 8.55	-	54.7; 8.68	53.4; 9.7	-	55.7; 9.02	51.8; 9.95	0.03
Fatigue ***	16.9; 3.86	15.3; 2.62	0.004	17; 3.67	15.6; 3.5	0.005	17.2; 3.71	16.6; 3.71	-
Monotony ***	17.9; 2.35	17.1; 2.38	0.046	18.3; 2.76	18.1; 2.53	-	18.1; 2.89	18.7; 2.48	-
Satiety ***	18.9; 4.2	17; 3.38	0.004	18.2; 4.36	17.8; 3.52	-	19.8; 4.85	18.3; 4.64	-
Stress ***	17.8; 3.6	18.6; 3.42	-	17.5; 3.07	18; 2.73	-	17.7; 3.24	19.7; 2.42	0.001
Depression ****	1.1; 1.69	2.4; 2.18	0.001	0.3; 0.79	2.3; 2.29	0.001	0.2; 0.52	2.4; 2.71	0.001
Anxiety ****	1.7; 3.07	1.8; 1.82	0.05	0.4; 1.04	1.7; 1.67	0.001	0.2; 0.64	1.9; 1.7	0.001
Stress ****	2.1; 3.11	3.9; 2.66	0.001	1.2; 1.81	3.8; 2.97	0.001	0.6; 1.59	4.2; 3.18	0.001

Note: K—subdivision of the village of Karpogory, Ya—subdivision of the village of Yasny; * G.A. Aminev’s coefficients for M. Luscher’s test, ** Status scale, *** DASRWC; **** DASS-21.

**Table 5 healthcare-13-02260-t005:** Labor productivity of loggers of two divisions at the beginning, middle, and end of the fly-in period.

Condition Parameter, Methodology	Beginning of Fly-In Period (M; SD)			Middle of Fly-In Period (M; SD)			End of Fly-In Period (M; SD)		
	K N = 63	Ya N = 98	*p*	K N = 63	Ya N = 98	*p*	K N = 63	Ya N = 98	*p*
Volume of timber per shift	144.2; 52.92	144.9; 38.75	-	97.7; 52.23	174.9; 48.87	0.001	145.5; 37.4	195.7; 60	0.003

Note: K—subdivision of the village of Karpogory, Ya—subdivision of the village of Yasny.

## Data Availability

Certificate of registration of the database 2021621449, 7 May 2021. Application No. 2021621307 dated 24 June 2021. Dynamic study of the functional states of workers of a logging enterprise in the Far North during a rotational race.

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
