# Peer review of "The Relationship Between Psychophysiological and Psychological Parameters of Job Stress and Working Capacity of Loggers During the Fly-In Period"

_healthcare, 2025, doi:10.3390/healthcare13182260_

Round 1

Reviewer 1 Report

Comments and Suggestions for Authors

Abstract

  • The abstract provides a clear overview of the study aim and summarizes key findings, but it lacks quantitative details on main results such as effect sizes or practical implications, which limits its informative value for OHS stakeholders.
  • It efficiently sets the context for the research; however, minor language ambiguities and some lengthy sentences impact its conciseness, and the linkage to journal scope (“Healthcare”) could be made more explicit by referencing broader healthcare significance.

Introduction (Background)

  • The introduction thoroughly details the occupational risks specific to logging in Arctic regions, directly aligning with OHS concerns and emphasizing the study’s practical relevance.
  • It could benefit from a more focused linkage to existing gaps in the occupational health literature, as portions remain somewhat descriptive rather than critically analytical.
  • The hypotheses are clearly stated, but greater explanation of their direct relevance to healthcare practice would strengthen suitability for the journal.

Methods

  • The longitudinal design, inclusion of validated psychological and physiological measurement tools, and statistical rigor (e.g., Bonferroni-Holm correction) enhance scientific validity and reliability.
  • The method section is comprehensive, but some operational definitions (e.g., specific cutoffs for assessment scales) and potential sources of bias (e.g., selection criteria, blinding) are not thoroughly addressed.
  • Practical implementation of multiple diagnostic devices is a strength, although the absence of external region comparison limits generalizability to other OHS contexts.

Results

  • The results are structured clearly and comparative statistics are well-presented, supporting scientific validity and direct applicability to OHS.
  • While tables provide detailed quantitative insights, interpretation within the body text would benefit from greater emphasis on practical significance rather than primarily statistical outcomes.
  • Identification of statistically significant differences between team characteristics and stress parameters is valuable, but critical discussion on causality and confounding factors is limited.

Discussion

  • The discussion contextualizes findings within the international literature, highlighting practical implications for shift work management and mental health interventions in logging.
  • It critically evaluates underlying mechanisms of stress and adaptation, but does not sufficiently address limitations regarding the regional focus and occupational group selection.
  • The section demonstrates strong coherence but occasionally reiterates previous results rather than expanding on practical strategies for OHS enhancement.

Conclusion

  • Conclusions are generally concise and actionable, emphasizing the importance of comprehensive monitoring and intervention for worker health and longevity.
  • The claim regarding broader applicability of monitoring methods could be better substantiated for diverse OHS environments.
  • The specific relevance for journal readership (“Healthcare”) would benefit from a clearer linkage to health system policies or practices.

References

  • The reference list is sufficiently up-to-date and diverse, citing relevant peer-reviewed sources and international research on occupational stress.
  • A few sources are regional or internal reports, which may not be globally accessible, slightly limiting reproducibility and verification.
  • There is an appropriate balance between empirical studies and foundational texts, though citation consistency and format would benefit from minor editorial revision.

Final Opinion

Major revisions required.
While the manuscript presents scientifically sound and practically valuable research that is largely suitable for publication in Healthcare, critical improvements are needed to strengthen clarity, enhance generalizability, bolster practical implications for OHS, and address limitations in sampling and comparative scope. Enhanced editing for conciseness and linkage to broader healthcare relevance is advised.

Comments on the Quality of English Language

Dear Authors,

Thank you for submitting your valuable manuscript entitled "The relationship between psychophysiological and psychological parameters of job stress and working capacity of loggers in the north at the beginning, middle and end of the shift period."

Your study demonstrates an appropriate research design and employs well-established methods. The results are clearly presented and supported by the data, and relevant references have been properly cited throughout the manuscript.

However, to enhance the clarity and overall quality of the paper, we recommend the following revisions:

  • Improve the presentation of the English language, particularly in complex sections, to ensure clearer and more fluent expression of ideas.

  • Provide additional details in the Methods section to facilitate reproducibility and to strengthen methodological transparency.

We believe that addressing these points will significantly improve the manuscript’s readability and impact.

Thank you for your contribution, and we look forward to receiving your revised manuscript.

Best regards,

Author Response

Dear Reviewer,

We would like to extend our sincerest gratitude for your time and effort in reviewing our manuscript and for providing these insightful and constructive comments. Your feedback has been invaluable in helping us improve the quality and clarity of our work.

We have carefully considered all your points and have revised the manuscript accordingly. All changes have been implemented in the text and are highlighted in green for your convenience, making them easy to track.

Point-by-point, we have addressed each of your comments below. Please find our responses detailed beneath each remark, where we explain the revisions made.

Comment 1. Improve the presentation of the English language, particularly in complex sections, to ensure clearer and more fluent expression of ideas.

Answer 1: changed the text of the manuscript.

Comment 2: Provide additional details in the Methods section to facilitate reproducibility and to strengthen methodological transparency.

Answer 2: added descriptions of the methods:

lines 233-236: The diagnostic program was developed over a long period of time in the course of a number of studies [25; 26; 31], and it is described in detail and tested in other studies, and includes a multi-component assessment of conditions using hardware, survey and pro-jective methods for the most complete description of job stress and working capacity.

lines 244-258: The VCM methodology distinguishes six levels of functional state:

Optimal (6): Characterized by the ideal tension of regulatory systems required to maintain the body's active equilibrium with the external environment.

Close to Optimal (5): A state of moderate tension where the body must utilize addi-tional functional reserves to adapt to environmental conditions.

Permissible (4): A state of pronounced tension, involving the active mobilization of protective mechanisms.

Maximum Permissible (3): A state of regulatory system overstrain, marked by the insufficiency of protective and adaptive mechanisms and an inadequate response to en-vironmental factors.

Negative (2): A state of regulatory system exhaustion, where the activity of control mechanisms declines. Individuals in this state must suppress their need for rest by will-power.

Critical (1): Characterized by functional disorders and a powerful, compelling need for rest as the restorative systems are activated.

lines 315-347: the method of differentiated assessment of states of reduced performance (DASRWC) by A.B. Leonova, S.B. Velichkovskaya. The method is an adaptation of the BMSII test by H. Plath and R. Richter [40; 41]. The test comprises 40 questions. Responses are assessed on a four-point scale (where 1 is "almost never" and 4 is "almost always") to identify four severity levels of symptoms for the following mental states: monotony, satiety, stress, and fatigue.

Fatigue is characterized by exhaustion and impaired coordination of core cognitive processes. It results from prolonged and intense workload under a dominant motivation to cease work and rest. Score interpretation: ≤15 (low); 16–25 (moderate); 26–31 (pro-nounced); >31 (high).

Monotony is a state of reduced conscious control over task performance. It occurs during monotonous work involving frequent repetition of stereotypical actions in an impoverished external environment, and is accompanied by feelings of boredom, drowsiness, and a strong desire to change activities. Score interpretation: ≤15 (low); 16–25 (moderate); 26–30 (pronounced); >30 (high).

Satiety (or mental satiety) is a state of aversion to overly simple, subjectively unin-teresting, or meaningless work. It manifests as a strong urge to either stop the activity altogether or to introduce variety into the established routine. Score interpretation: ≤16 (low); 17–24 (moderate); 25–30 (pronounced); >30 (high).

Stress (or tension) is a state of heightened mobilization of psychological and ener-getic resources in response to increased task difficulty or subjective significance. Score interpretation: ≤16 (low); 17–24 (moderate); 25–30 (severe); >30 (high).

         Depression, Anxiety, and Stress Scale (DASS-21) by S. Lovibond and P. Lov-ibond, adapted by A.A. Zolotareva [42; 43; 44; 45] consists of 21 statements to be rated on a 4-point scale from never to almost always. The depression scale measures such mental states and experiences as dysphoria, hopelessness, devaluation of life, self-abasement, apathy (1-8 points indicate experiences of depressive spectrum symptoms from mild de-pression (subdepression) to moderate; more than 8 points indicate high severity of de-pressive state). The anxiety scale assesses vegetative arousal, situational anxiety and subjective experience of experiencing anxiety states (1-7 points - from mild to moderate anxiety; more than 7 points - high severity). The stress scale is sensitive to chronic non-specific arousal and diagnoses difficulties with relaxation, nervous arousal and irri-tability (2-10 points from mild to moderate stress, more than 10 points - severe stress).

Comment 3: Abstract

The abstract provides a clear overview of the study aim and summarizes key findings, but it lacks quantitative details on main results such as effect sizes or practical implications, which limits its informative value for OHS stakeholders.

It efficiently sets the context for the research; however, minor language ambiguities and some lengthy sentences impact its conciseness, and the linkage to journal scope (“Healthcare”) could be made more explicit by referencing broader healthcare significance.

Answer 3: The abstract has been completely changed.

Comment 4: Introduction (Background)

The introduction thoroughly details the occupational risks specific to logging in Arctic regions, directly aligning with OHS concerns and emphasizing the study’s practical relevance.

It could benefit from a more focused linkage to existing gaps in the occupational health literature, as portions remain somewhat descriptive rather than critically analytical.

The hypotheses are clearly stated, but greater explanation of their direct relevance to healthcare practice would strengthen suitability for the journal.

Answer 4: The following information has been added. Lines 124-133 и 153-172:

This study investigated the relationship between psychophysiologi-cal/psychological parameters of job stress and the working capacity of loggers at dif-ferent stages of the fly-in period. Its findings hold significant value for public health and occupational medicine. By identifying the most demanding stages of the fly-in cy-cle, targeted preventive interventions can be implemented. These may include en-hanced monitoring of stress and working capacity, along with psychological support.

The combination of psychophysiological and psychological markers we identified can form the basis for diagnostic panels or screening programs. This will enable on-site medical personnel (paramedics and physicians) to detect at-risk workers earlier, before the clinical onset of chronic stress, cardiovascular pathologies (e.g., arterial hyperten-sion), and neuropsychiatric disorders.

As logging is a high-risk industry for injuries and accidents—often caused by hu-man factors like fatigue, inattention, and emotional instability—applying our findings to the occupational safety system will help:

         reduce industrial injuries through proactive management of worker fatigue and stress;

         minimize economic losses related to absenteeism, sick pay, and employee turnover;

         increase productivity by maintaining an optimal level of worker perfor-mance throughout the fly-in period.

Comment 5: Discussion

The discussion contextualizes findings within the international literature, highlighting practical implications for shift work management and mental health interventions in logging.

It critically evaluates underlying mechanisms of stress and adaptation, but does not sufficiently address limitations regarding the regional focus and occupational group selection.

The section demonstrates strong coherence but occasionally reiterates previous results rather than expanding on practical strategies for OHS enhancement.

Answer 5: additions have been made to the discussion of the results. Lines 662-704 и lines 730-743:

The findings of the study can be adapted to a wide range of occupations charac-terized by similar production factors (FIFO work in group isolation conditions, for example, the oil and gas industry, the mining industry). This opens up significant pro-spects for the development of universal protocols in the public health system.

Based on the findings of this study, the following practical recommendations are proposed to enhance performance and safeguard the health of loggers:

  1. For the occupational safety system and employers:

- Enhance safety protocols during critical periods: Strengthen oversight of occu-pational safety compliance during the beginning and end of the fly-in period. Tempo-rarily restrict or increase supervision for the most hazardous tasks requiring high con-centration (e.g., felling trees, working near machinery) during these high-risk phases.

- Implement targeted psychophysiological relief programs:

Beginning of fly-in period: Focus on accelerated adaptation techniques and anxie-ty reduction (e.g., guided breathing exercises, adaptation briefings).

End of fly-in period: Introduce controlled recovery programs to combat cumula-tive fatigue (e.g., short relaxation sessions, facilitating quality rest).

-          Modify work-rest schedules: consider adjusting shift structures, for example by reducing shift duration to 11 hours and increasing rest time. This could be achieved by logistical improvements, such as ensuring transportation from the shift camp to the work site takes no more than 20-30 minutes.

  1. For the medical support and public health systems:

- Implement a functional state screening system:

Utilize the DASRWC questionnaire as the primary tool for weekly screening of all personnel, as its sufficiency and ease of use have been validated.

Conduct more in-depth psychophysiological testing (using hardware methods) selectively for workers in the "risk group" (as identified by DASRWC results) or during periodic medical examinations. This targeted approach optimizes the medical unit's resources.

-          Develop early intervention protocols for specialists: create standardized work protocols for psychologists and medical staff:

Beginning of fly-in period: techniques to reduce adaptation stress and foster team cohesion.

End of fly-in period: methods for providing emotional support and cognitive mo-bilization to ensure work is completed safely.

-          Establish professional longevity programs: use the data to create individualized rehabilitation programs for the inter-shift period. These programs should aim to compensate for the identified loads (e.g., cardiovascular rehabilitation, stress management programs).

-          Create educational materials for workers: develop clear guides and memos that explain the typical dynamics of their condition during a shift, teach self-monitoring techniques, and emphasize the importance of seeking help during critical periods.

The obtained results yield practical recommendations for the targeted manage-ment of the socio-psychological climate in logging teams as a resource for maintaining health and performance:

         form stable work teams, considering employee preferences during staffing to en-hance work comfort;

         conduct an annual assessment of the socio-psychological climate, including em-ployee feedback on improving intra-team and management interaction;

         incorporate socio-psychological climate metrics into the key performance indi-cators (KPIs) for foremen and line managers to incentivize their focus on this issue;

         hold regular short meetings (e.g., toolbox talks). Use these not only for safety briefings and task, but also to openly discuss difficulties, assign tasks considering em-ployee input, and express gratitude for good work. This fosters a greater sense of in-volvement and fairness.

Comment 6: Conclusion

Conclusions are generally concise and actionable, emphasizing the importance of comprehensive monitoring and intervention for worker health and longevity.

The claim regarding broader applicability of monitoring methods could be better substantiated for diverse OHS environments.

The specific relevance for journal readership (“Healthcare”) would benefit from a clearer linkage to health system policies or practices

Answer 6: Conclusions have been reformulated. Lines 775-809.

We sincerely appreciate your time and the encouraging feedback on our research.

With our best wishes, authors

Reviewer 2 Report

Comments and Suggestions for Authors

thank you very much. it seems novel work. I have listed my recommendations:

1-)the title can be abbreviated. the title seems slightly unusual for a research paper.

2-)in your title ''in the north'' seems unclear. 

3-)please highlight the novelty of the study in the abstract. also, please mention the gap in the literature.

4-)in the title, the beginning middle and end of the shift period sounds slightly unusual as well. please consider writing it concisely.

5-)please mention limitations of the study related to methodology of your work in the limitations section.

6-)be sure that the reference is precise and the information is universal.

 Long working hours (more than 12) and work requirements increase the likelihood 49
of fatigue and stress in loggers [9]. 

7-)please be sure the references are precise and necessary.

Literature data show that working conditions and pro- 50
fessional workloads have a negative impact on the functional state of loggers: there is an 51
increase in stress, anxiety, as well as tension in different parts of the body (back, neck, etc.) 52
[8; 10; 11].

8-)please improve the sentence:

Our previous studies have shown differences in 90
the dynamics of performance and stress, measured using biochemical, psychophysiolog- 91
ical and psychological methods, in shift workers of oil and gas production enterprises in 92
the Arctic [24] and logging workers in the Far North [23]

9-)be sure the following is precisely referring the methods implemented in your study:

beginning, middle and end of the shift period

10-)this sentence can be paraphrased to sound academic:

Special hypothesis: dynamic shifts in the functional states of loggers during the shift 122
period are positively associated with positive socio-psychological characteristics of the 123
logging team.

11-)please give more information about job positions.

Author Response

Dear Reviewer,

We would like to extend our sincerest gratitude for your time and effort in reviewing our manuscript and for providing these insightful and constructive comments. Your feedback has been invaluable in helping us improve the quality and clarity of our work.

We have carefully considered all your points and have revised the manuscript accordingly. All changes have been implemented in the text and are highlighted in green for your convenience, making them easy to track.

Point-by-point, we have addressed each of your comments below. Please find our responses detailed beneath each remark, where we explain the revisions made.

Comment 1. the title can be abbreviated. the title seems slightly unusual for a research paper.

Answer 1: The title has been shortened. The relationship between psychophysiological and psycholog-ical parameters of job stress and working capacity of loggers during the fly-in period

Comment 2. in your title ''in the north'' seems unclear.

Answer 2: The title e has been shortened. The relationship between psychophysiological and psycholog-ical parameters of job stress and working capacity of loggers during the fly-in period

Comment 3. please highlight the novelty of the study in the abstract. also, please mention the gap in the literature.

Answer 3: Added: Background: Scientific research on fly-in/fly-out (FIFO) workers has identified a gap in under-standing the dynamics of job stress parameters among forest workers throughout the shift cycle. Lines 11-13.

Comment 4. in the title, the beginning middle and end of the shift period sounds slightly unusual as well. please consider writing it concisely.

Answer 4: The title e has been shortened. Also, the key term shift method has been changed to fly-in-fly-out (FIFO) method, and shift period to fly-in period. The introduction provides a detailed description of what the FIFO method is. Lines 55-67:

Due to the remoteness of forest plots from populated areas (more than 100 km) and their constant movement through forest areas, logging enterprises use the shift work organization method or FIFO [8]. A fly-in-fly-out basis (FIFO) is defined as “any employment in which work is so isolated from workers' homes that they are provided with food and housing in the workplace, and schedules are created whereby workers spend a fixed number of days at the field followed by a fixed number of days at home ”[9, p.2]. In FIFO work in logging, the shift duration is usually 12 hours and employees work daily for 15 days without days off, followed by a 15-day rest fly-out period, and then another 15-day fly-in period. This work involves 12-hour day and night shifts, so in the middle of the fly-in period there is a changeover, when day shift workers switch to night shift workers, and vice versa. The FIFO method is widely used in the extractive industries (oil and gas, diamond and mining, etc.), which extract minerals in remote areas with harsh climatic conditions of Arctic and High North [10].

Comment 5. please mention limitations of the study related to methodology of your work in the limitations section.

Answer 5: Added. Lines 750-752: The results obtained are based on the methods of interviews, questionnaires, psycho-physiological and psychological testing. The use of other tests and methods may allow us to refine the results obtained

Comment 6. be sure that the reference is precise and the information is universal.

 Long working hours (more than 12) and work requirements increase the likelihood 49

of fatigue and stress in loggers [9].

Answer 6: We have clarified that the link is correct.

Comment 7. please be sure the references are precise and necessary.

Literature data show that working conditions and pro- 50

fessional workloads have a negative impact on the functional state of loggers: there is an 51

increase in stress, anxiety, as well as tension in different parts of the body (back, neck, etc.) 52

[8; 10; 11].

Answer 7: The references are pertinent and well-chosen, providing a foundation for describing the detrimental impact of forestry work on occupational health and worker well-being.

Comment 8. please improve the sentence:

Our previous studies have shown differences in 90

the dynamics of performance and stress, measured using biochemical, psychophysiolog- 91

ical and psychological methods, in shift workers of oil and gas production enterprises in 92

the Arctic [24] and logging workers in the Far North [23]

Answer 8: The sentence has been corrected. Lines 113-116. Our previous studies have used biochemical, psychophysiological, and psycho-logical methods to identify differences in the dynamics of working capacity and stress between two worker groups: FIFO employees in the Arctic oil and gas sector [26] and logging workers in the Far North [25].

Comment 9. be sure the following is precisely referring the methods implemented in your study:

beginning, middle and end of the shift period.

Answer 9: Since this was a longitudinal study, describing the stages of the fly-in period and the number of measurements taken at each stage was important for understanding the study's design and its subsequent analysis.

Comment 10. this sentence can be paraphrased to sound academic:

Special hypothesis: dynamic shifts in the functional states of loggers during the shift 122

period are positively associated with positive socio-psychological characteristics of the 123

logging team.

Answer 10: The sentence has been corrected. Lines 150-152. The special hypothesis posits that positive socio-psychological characteristics within a logging team are positively associated with dynamic changes in work capacity and job stress parameters among workers during the FIFO rotation period.

Comment 11. please give more information about job positions.

Answer 11: Information added. Lines 216-225. The study focused on a primary professional group: operators of logging equipment. This group comprised several specialized roles:

Harvester and forwarder operators: these operators work in tandem. Harvester operators perform felling, limbing, and cross-cutting using automated systems. Forwarder operators are then responsible for collecting, loading, and transporting the timber to the roadside using a crane-manipulator, ensuring the harvester's continuous operation.

Support operators: this category included motor grader operators, who build and maintain the road network, and drivers who operate timber and dump trucks.

Technical maintenance: specialists such as welders were included for their role in performing on-site minor equipment repairs.

We sincerely appreciate your time and the encouraging feedback on our research.

With our best wishes, authors

Reviewer 3 Report

Comments and Suggestions for Authors

The manuscript addresses an important topic, but several aspects require clarification and improvement.

Section 1. Introduction

Numerous risk factors are listed together, without separation into physical, psychological, and organizational categories.

The research problem is not clearly stated. The discrepancy between psychological and psychophysiological parameters is mentioned but not framed as a research gap.

Prior studies on forestry and oil & gas workers are cited, but the rationale for comparing these industries is not explained.

The choice to study the summer period is noted, but its relevance relative to existing winter studies is not justified.

The objectives are presented, but their connection to the hypotheses is not made explicit.

The hypotheses are broad and do not indicate which variables are expected to correlate or differ at each shift stage.

Section 2. Materials and Methods

The procedure is described in detail, but the decision to merge the two sites into one sample is weakly supported. Differences in age, experience, and job structure (Table 1) could affect the results.

The sample size (n=47) is small, and no power calculation is reported. Representativeness is claimed but not demonstrated statistically.

Numerous psychophysiological and psychological tools are listed. The choice of these methods and their validation for logging or shift-work populations are not explained.

The grouping of days (1–3, 6–8, 11–13) is not justified with theoretical or empirical arguments

At line 178, the reference to Figure 1 does not correspond to its graphical representation.

Section 3. Results

Interpretative statements should not appear in this section and must be either moved to the Discussion section or eliminated (e.g., lines 318–320, 397–399, 422–426, 439–447).

Terms such as “favorable”, “above average”, or “acceptable” should be avoided. Only numerical results should remain here.

Correlation analysis should be limited to coefficients and significance; explanatory mechanisms should be moved to the Discussion section.

Productivity data should be reported descriptively, without linking to stress levels in this section.

Results from all psychological tests should be explicitly shown in tables or figures; some methods (e.g., Luscher coefficients) are not clearly reported.

Information on missing data, effect sizes, and variability should be added.

Section 4. Discussion

Descriptive repetition of Results section should be removed. Only interpretation and comparison with previous studies should remain.

Statements such as “workers had a close to optimal state and above average functional capabilities” (lines 452–454) duplicate Results section and should be moved there or deleted.

The discussion of summer versus winter productivity and accidents (lines 457–466) is relevant, but the link with the current study is not demonstrated. Authors should clarify whether their data confirm or contradict these references.

The section introduces mechanisms (compensatory reactions, orienting reaction with experience, lines 483–490) without sufficient statistical support. These should be presented more cautiously.

Hypotheses are said to be confirmed (lines 491–494, 523–527), but the basis is descriptive rather than analytical. Authors should show direct evidence supporting confirmation or partial confirmation.

The claim that pressure-based coefficients can be used for workplace feedback and interventions (lines 509–521) goes beyond the presented data. It should be presented as a potential application, rather than an established practice.

The part on team communication and isolation (lines 528–546) cites literature but is not linked back to this dataset. It should be shortened or explicitly related to findings.

The discussion does not explicitly state the study’s purpose, relevance, or contribution. Please clarify how the results advance knowledge on occupational risks for loggers and what practical measures may be derived.

Limitations are too general and should be quantified to possible effects on results (e.g., sample size, regional specificity, self-report bias). The discussion does not suggest how future work could address them.

Section 5. Conclusions

The Conclusions section repeats the Results section without stating the study’s contribution or practical implications. Please specify what new knowledge is provided and how it can be applied or extended.

Comments on the Quality of English Language

I prefer not to comment on the language quality and recommend that it be reviewed by someone with expertise.

Author Response

Dear Reviewer,

We would like to extend our sincerest gratitude for your time and effort in reviewing our manuscript and for providing these insightful and constructive comments. Your feedback has been invaluable in helping us improve the quality and clarity of our work.

We have carefully considered all your points and have revised the manuscript accordingly. All changes have been implemented in the text and are highlighted in green for your convenience, making them easy to track.

Point-by-point, we have addressed each of your comments below. Please find our responses detailed beneath each remark, where we explain the revisions made.

Comment 1. Numerous risk factors are listed together, without separation into physical, psychological, and organizational categories.

Answer 1: Added. Lines 33-37. Physical (low temperatures, noise, vibration) and production factors (heavy physical labor, risk of injury) in combination with organizational (disruption of biorhythms, in-tensity of workload, duration of shift period) and psychological (psychosocial stress, group isolation) factors create a unique set of professional risks for loggers.

The logic of the introduction presents first physical and professional factors, then organizational and psychological ones.

Comment 2. The research problem is not clearly stated. The discrepancy between psychological and psychophysiological parameters is mentioned but not framed as a research gap.

Answer 2: Added. Lines 109-112. Thus, in modern science there remains a significant gap in understanding the dynamics of the relationship between psychophysiological markers of job stress and working capacity level of workers in the logging industry at various stages of the fly-in period.

Comment 3. Prior studies on forestry and oil & gas workers are cited, but the rationale for comparing these industries is not explained.

Answer 3: Added definition of the fly-in-fly-out (FIFO) method and its application in the oil and gas production sector. Lines 55-67.

Due to the remoteness of forest plots from populated areas (more than 100 km) and their constant movement through forest areas, logging enterprises use the shift work organization method or FIFO [8]. A fly-in-fly-out basis (FIFO) is defined as “any employment in which work is so isolated from workers' homes that they are provided with food and housing in the workplace, and schedules are created whereby workers spend a fixed number of days at the field followed by a fixed number of days at home ”[9, p.2]. In FIFO work in logging, the shift duration is usually 12 hours and employees work daily for 15 days without days off, followed by a 15-day rest fly-out period, and then another 15-day fly-in period. This work involves 12-hour day and night shifts, so in the middle of the fly-in period there is a changeover, when day shift workers switch to night shift workers, and vice versa. The FIFO method is widely used in the extractive industries (oil and gas, diamond and mining, etc.), which extract minerals in remote areas with harsh climatic conditions of Arctic and High North [10].

Comment 4. The choice to study the summer period is noted, but its relevance relative to existing winter studies is not justified.

Answer 4: Added. Lines 124-129.

The research was conducted in the summer period due to the need to study the impact of a complex of specific factors on workers: maximum work intensity, abnormally high temperatures for the North, increased activity of blood-sucking insects (gnats) and the polar day, which can provoke a unique pattern of job stress and decreased working capacity that is not typical for other seasons.

Comment 5. The objectives are presented, but their connection to the hypotheses is not made explicit.

Answer 5: The wording of the objectives has been changed.

The aim of this study is to explore the complex interplay between psychologi-cal/psychophysiological markers of job stress and work capacity among FIFO loggers in the Arctic region. A key focus is to analyze how this interplay evolves during the fly-in cycle and is modulated by the socio-psychological climate of the work teams.

Comment 6. The hypotheses are broad and do not indicate which variables are expected to correlate or differ at each shift stage.

Answer 6: The hypotheses were left unspecified due to the study's exploratory nature, as its purpose was not to test predetermined points.

Comment 7. The procedure is described in detail, but the decision to merge the two sites into one sample is weakly supported. Differences in age, experience, and job structure (Table 1) could affect the results.

Answer 7: Added. Lines 201-213.

This longitudinal study was conducted via expeditions, where the researchers re-sided on-site for the entire fly-in period to conduct twice-daily monitoring of the log-gers' conditions. To maximize the sample size, the largest available plots were selected (approximately 30 loggers each). While a smaller plot of 12 loggers was also included, it required the same extensive expeditionary resources. The professional activities and organizational parameters were identical across both samples; the only geographical difference was their 30 km separation, as season and weather conditions were the same. Consequently, the samples were considered equivalent and were pooled to in-crease the overall sample size. The only objective, uncontrolled differences between the groups were in age and work experience. The study's sufficient statistical power is supported by its longitudinal design and the high number of repeated measurements. The final compared sample sizes for stress and performance measurements were 161 (beginning), 194 (middle), and 164 (end of the fly-in period).

Comment 8. The sample size (n=47) is small, and no power calculation is reported. Representativeness is claimed but not demonstrated statistically.

Answer 8: Added. Lines 201-213.

This longitudinal study was conducted via expeditions, where the researchers re-sided on-site for the entire fly-in period to conduct twice-daily monitoring of the log-gers' conditions. To maximize the sample size, the largest available plots were selected (approximately 30 loggers each). While a smaller plot of 12 loggers was also included, it required the same extensive expeditionary resources. The professional activities and organizational parameters were identical across both samples; the only geographical difference was their 30 km separation, as season and weather conditions were the same. Consequently, the samples were considered equivalent and were pooled to in-crease the overall sample size. The only objective, uncontrolled differences between the groups were in age and work experience. The study's sufficient statistical power is supported by its longitudinal design and the high number of repeated measurements. The final compared sample sizes for stress and performance measurements were 161 (beginning), 194 (middle), and 164 (end of the fly-in period).

Comment 9. Numerous psychophysiological and psychological tools are listed. The choice of these methods and their validation for logging or shift-work populations are not explained.

Answer 9: Added. Lines 233-236.

The diagnostic program was developed over a long period of time in the course of a number of studies [25; 26; 31], and it is described in detail and tested in other studies, and includes a multi-component assessment of conditions using hardware, survey and pro-jective methods for the most complete description of job stress and working capacity.

Comment 10. The grouping of days (1–3, 6–8, 11–13) is not justified with theoretical or empirical arguments

Answer 10: Added. Lines 406-410.

The introduction provides data on the previous expedition, where it was established that the most stressful days of the fly-in period are the beginning (1-3 days), the end (11-13) and the middle (6-8 days) due to the shift change. These data are also on other samples of FIFO workers in the oil and gas production industry in the Arctic [26].

Comment 11. At line 178, the reference to Figure 1 does not correspond to its graphical representation.

Answer 11: Corrected. Link removed.

Comment 12. Interpretative statements should not appear in this section and must be either moved to the Discussion section or eliminated (e.g., lines 318–320, 397–399, 422–426, 439–447).

Answer 12: To enhance clarity, brief interpretations have been incorporated into the results section. Furthermore, in response to feedback from other reviewers, we have expanded the analysis to better elucidate the obtained relationships and differences. The primary interpretation and synthesis of these findings have been moved to the Discussion section.

Comment 13. Terms such as “favorable”, “above average”, or “acceptable” should be avoided. Only numerical results should remain here.

Answer 13: The applied methods required the classification of indicators into distinct levels, each with a designated nomenclature. Additional information elucidating these levels has been incorporated into the manuscript. Lines 243-258 and 289-347.

Comment 14. Correlation analysis should be limited to coefficients and significance; explanatory mechanisms should be moved to the Discussion section.

Answer 14: While the Discussion section explores the most significant and thought-provoking connections, more straightforward correlations were briefly addressed in the Results section to prevent unnecessary repetition and enhance the narrative flow.

Comment 15. Productivity data should be reported descriptively, without linking to stress levels in this section.

Answer 15: Moved to a separate table 5.

Comment 16. Results from all psychological tests should be explicitly shown in tables or figures; some methods (e.g., Luscher coefficients) are not clearly reported. Information on missing data, effect sizes, and variability should be added.

Answer 16: Additions were made to Table 4.

Comment 17. Descriptive repetition of Results section should be removed. Only interpretation and comparison with previous studies should remain.

Answer 17: We appreciate this comment. In response, we performed a thorough review of the Discussion. Our goal was to make the section more concise. We identified that the clarifications regarding the matching characteristics were fundamental to our argument, so we have refined and retained them for greater impact.

Comment 18. Statements such as “workers had a close to optimal state and above average functional capabilities” (lines 452–454) duplicate Results section and should be moved there or deleted.

Answer 18: This phrase is intended to illustrate the manifestation of these parameters and integrate them into the broader discussion.

Comment 19. The discussion of summer versus winter productivity and accidents (lines 457–466) is relevant, but the link with the current study is not demonstrated. Authors should clarify whether their data confirm or contradict these references.

Answer 19: Demonstrating these connections in the discussion is intended to enhance the practical significance of the data obtained and to enable the obtained data to be used in practice.

Comment 20. The section introduces mechanisms (compensatory reactions, orienting reaction with experience, lines 483–490) without sufficient statistical support. These should be presented more cautiously.

Answer 20: We thank the reviewer for this important comment. We agree that these mechanisms should be presented more cautiously. We have revised the section to clarify that the proposed mechanisms are a hypothetical framework based on the cited literature and our qualitative observations, and that they require further validation with dedicated statistical analysis in future studies.

Comment 21. Hypotheses are said to be confirmed (lines 491–494, 523–527), but the basis is descriptive rather than analytical. Authors should show direct evidence supporting confirmation or partial confirmation.

Answer 21: At this stage, the descriptive nature of our conclusions cannot yet be replaced by an analytical one, as the underlying mechanisms of the discovered connections require further investigation. This work represents an initial foray into this area, and we hope to collaborate with physiologists in the future to elucidate these mechanisms.

Comment 22. The claim that pressure-based coefficients can be used for workplace feedback and interventions (lines 509–521) goes beyond the presented data. It should be presented as a potential application, rather than an established practice.

Answer 22: This is proposed as a recommendation. Current industry practice relies on an arbitrary set of disparate indicators, which often leads to an inadequate assessment of a worker's condition. We consider it our mission to develop simple yet accurate methods for assessing workers in remote forest plots.

Comment 23. The part on team communication and isolation (lines 528–546) cites literature but is not linked back to this dataset. It should be shortened or explicitly related to findings.

Answer 23: Added. Lines 708-709. Explanations included.

Comment 24. The discussion does not explicitly state the study’s purpose, relevance, or contribution. Please clarify how the results advance knowledge on occupational risks for loggers and what practical measures may be derived.

Answer 24: Added practical recommendations. Lines 662-704 and 730-743

Comment 25. Limitations are too general and should be quantified to possible effects on results (e.g., sample size, regional specificity, self-report bias). The discussion does not suggest how future work could address them.

Answer 25: Changed and added. 744-774

Comment 26. The Conclusions section repeats the Results section without stating the study’s contribution or practical implications. Please specify what new knowledge is provided and how it can be applied or extended.

Answer 26: Conclusions have been changed.

This study established statistically significant relationships between the psychological and psychophysiological parameters of functional states in Arctic loggers across a 15-day fly-in period. The most significant findings are summarized below.

Dynamics of functional state

The adaptation (beginning) and fatigue (end) phases of the fly-in period were identi-fied as the most stressful. During these stages, assessments of functional state were more consistent but less favorable compared to the more stable middle phase. Notably, we observed a psychological mobilization effect at the start of the period, where sub-jective comfort was maintained despite physiological strain. Conversely, by the end, workers demonstrated a relative maintenance of functional capabilities against a backdrop of accumulating fatigue.

Practical application of monitoring

A strong correlation was found between objective (equipment-based) parameters and the subjective DASRWC questionnaire. This validates the use of this concise question-naire as a standalone, efficient tool for routine monitoring of functional states in re-mote industrial settings, enabling timely psychological or physiological intervention.

Impact of team socio-psychological climate

The socio-psychological climate of a team was a critical factor in worker well-being:

Positive climate teams: workers in these teams exhibited a more favorable functional state throughout the fly-in period, characterized by moderate fatigue and stress, aver-age to above-average operational performance, and high adaptive and communicative abilities.

Moderate climate teams: workers in these teams reported higher stress, lower opera-tional performance, and only an acceptable functional state. This appears linked to their lower adaptive abilities and communicative potential.

Performance outcomes

A notable finding was that teams with a moderate socio-psychological climate achieved higher timber harvest volumes mid- and end-period. This may be driven by a stronger adherence to organizational norms and a heightened focus on meeting per-formance targets, even at a higher personal cost of stress.

The functional state of loggers is dynamic and significantly influenced by the phase of the fly-in period and the quality of their team environment. Implementing simplified monitoring tools like the DASRWC questionnaire can help safeguard worker health and promote professional longevity in this demanding industry.

We sincerely appreciate your time and the encouraging feedback on our research.

With our best wishes, authors

Round 2

Reviewer 1 Report

Comments and Suggestions for Authors

Dear Authors ,

Thank you for your diligent work in addressing the revisions.
The modifications have been thoroughly addressed and are now approved.

Wishing you continued success in your future endeavors.
Reviewer

Reviewer 2 Report

Comments and Suggestions for Authors

I accept

Reviewer 3 Report

Comments and Suggestions for Authors

Thank you for responding to my suggestions.